# Investigation of the Effect of the Auxin Antagonist PEO-IAA on Cannabinoid Gene Expression and Content in *Cannabis sativa* L. Plants under In Vitro Conditions

**DOI:** 10.3390/plants12081664

**Published:** 2023-04-15

**Authors:** Josef Baltazar Šenkyřík, Tereza Křivánková, Dominika Kaczorová, Nikola Štefelová

**Affiliations:** 1Department of Botany, Faculty of Science, Palacký University Olomouc, 78371 Olomouc, Czech Republic; 2Czech Advanced Technology and Research Institute, Palacký University Olomouc, 78371 Olomouc, Czech Republic; 3Centre of the Region Haná for Biotechnological and Agricultural Research, Department of Genetic Resources for Vegetables, Medicinal and Special Plants, Crop Research Institute, 78371 Olomouc, Czech Republic; 4Department of Biochemistry, Faculty of Science, Palacký University, 78371 Olomouc, Czech Republic

**Keywords:** *Cannabis sativa*, in vitro, shoot propagation, auxin antagonist, PEO-IAA, cannabinoids, qRT-PCR, THC, CBD, CBC

## Abstract

The in vitro shoot propagation of *Cannabis sativa* L. is an emerging research area for large-scale plant material production. However, how in vitro conditions influence the genetic stability of maintained material, as well as whether changes in the concentration and composition of secondary metabolites can be expected are aspects that need to be better understood. These features are essential for the standardised production of medicinal cannabis. This work aimed to find out whether the presence of the auxin antagonist α-(2-oxo-2-phenylethyl)-1H-indole-3-acetic acid (PEO-IAA) in the culture media influenced the relative gene expression (RGE) of the genes of interest (OAC, CBCA, CBDA, THCA) and the concentrations of studied cannabinoids (CBCA, CBDA, CBC, ∆^9^-THCA, and ∆^9^-THC). Two *C. sativa* cultivars, ‘USO-31’ and ‘Tatanka Pure CBD’, were cultivated by in vitro conditions with PEO-IAA presence and then analysed. The RT-qPCR results indicated that even though some changes in the RGE profiles could be observed, no differences were statistically significant compared with the control variant. The results of the phytochemical analyses demonstrate that although there were some differences from the control variant, only the cultivar ‘Tatanka Pure CBD’ showed a statistically significant increase (at a statistical significance level α = 0.05) in the concentration of the cannabinoid CBDA. In conclusion, it would appear that using PEO-IAA in the culture medium is a suitable approach to improve in vitro cannabis multiplication.

## 1. Introduction

*Cannabis sativa* L. is an annual dioecious herb with a centre of origin in the north-eastern Tibetan Plateau, from where it was later dispersed to the west (Europe) and east (China) [1]. Cannabis has been widely cultivated due to its industrial [2], nutritional [3], medicinal, and recreational [4] potentials. From regulatory and application perspectives, cannabis plants are categorized based on the level of Δ9-tetrahydrocannabinol (THC), one of the most important phytocannabinoids [5]. Plants are generally classified and regulated as industrial hemp if they contain less than 0.3% THC in the dried female inflorescence (this level varies by country; in the Czech Republic, since the year 2022, it is a level of 1%) or otherwise as drug-type, if the percentage is greater than this threshold [6]. Nevertheless, its phytochemical composition has made it equally important in traditional medicine [7]. The pharmaceutically most attractive secondary metabolites are cannabinoids, which belong to the unique class of terpene phenolic compounds that mainly accumulate in the glandular trichomes of the female inflorescences [8]. Cannabis contains more than 100 phytocannabinoids, of which the most abundant and significant are psychoactive Δ^9^-tetrahydrocannabinol (∆^9^-THC), non-psychoactive cannabidiol (CBD), and cannabichromene (CBC) with a low affinity for the endocannabinoid receptors [9]. The current surge of interest in this crop, not purely in the scientific community, is driven by an increasing number of countries relaxing restrictive approaches to cannabis research, commercial cultivation, and the sale of dried cannabis inflorescences, extracts, and edible medical or industrial products [10].

Phytocannabinoids are predominantly biosynthesised in the glandular trichomes of female cannabis plants, which are highly concentrated in the inflorescence. However, glandular trichomes are also partially located on leaves, where phytocannabinoids may also be present in lower amounts compared to flowers [11,12]. The occurrence is used for plant chemotypic classification in the early stage of vegetative growth [13]. Cannabinoid compounds result from the coupling of two biosynthetic pathways, the polyketide and 2-C-methyl-D-erythritol 4-phosphate (MEP) (Figure 1). Cannabigerolic acid (CBGA) is the biosynthetic precursor of all pentyl cannabinoids and is the product of the enzymatic reaction of olivetolic acid and geranyl pyrophosphate. Then, three major phytocannabinoids are produced from CBGA: cannabidiolic acid (CBDA), ∆^9^-tetrahydrocannabinolic acid (∆^9^-THCA), and cannabichromenic acid (CBCA), each reaction being catalysed via the respective oxidocyclases (CBDA synthase (CBDAS), THCA synthase (THCAS), CBCA synthase (CBCAS)). Their neutral homologs (CBD, ∆^9^-THC, CBC) are products of acid decarboxylation that occurs during heating, light exposure, storage, and maturation of plant material [12,14].

The inheritance of THCA and CBDA synthases is still unclear. Generally, two theories of the genetic origin of the enzymes exist, a Mendelian inheritance model and a multilocus model. The first model assumes that two codominant alleles at a single locus encode *CBDAS* and *THCAS*. The homozygous loci lead to the prevalence of CBD or ∆^9^-THC. In the case of two different alleles, the plant produces both major cannabinoids [15]. However, this model does not resolve the presence of CBC. Recent genomic studies preferentially support an alternative inheritance theory that *THCAS* and *CBDAS* are two different genes in a nearby genomic region [16].

The progress in large-scale plant material production, breeding programs, and biotechnological research are limited by the ability to perform robust and reproducible in vitro cultivation methods. It is problematic to maintain elite cultivars with seeds. Therefore, the clonal collections (mainly for medicinal genotypes) are advantageous due to the improved genetic conservation of lines for their pharmacological properties [17]. In vitro plant tissue culture techniques are a fundamental approach for plant propagation. In addition, they represent a valuable tool for studying plant developmental changes at the physiological, morphological, and molecular levels. However, multiplication and rooting still represent, to some extent, a bottleneck for trouble-free in vitro cultivation of cannabis. Although the rate of new scientific publications on the in vitro cultivation of cannabis is accelerating, adapting protocols for the micropropagation of different genotypes remains challenging [18,19,20,21]. Generally, the successful cultivation of plants under in vitro conditions highly depends on the genotype, the explant type, the composition of the medium, light intensity and quality, carbohydrate source, and, above all, the presence of phytohormones in the medium [22,23,24]. Various phytohormones have been tested to improve cannabis micropropagation protocols, such as thidiazuron (TDZ) [17,25], 1-naphthalene acetic acid (NAA), and 6-benzylaminopurine (BAP) [26] for shoot proliferation and regeneration, and indole-3-butyric acid (IBA) [21] and NAA [27] for rooting induction, while Lata et al. (2016) tested *meta*-topolin (mT) for a shoot and root induction from nodal segments [18]. Large-scale micropropagation, genetic conservation, and the development of plant biotechnologies for advanced new plant breeding technologies (NPBTs) might be promising tools for future research and commercial production of cannabis, although the plant is still understudied [10,28,29,30].

One alternative to increase the shoot multiplication rate under in vitro conditions is to disrupt apical dominance, which is very often prominent in cannabis (usually caused by very high endogenous auxin (Aux) concentrations). One of the substances suitable for disrupting shoot apical dominance is PEO-IAA, a novel molecule exhibiting strong anti-auxin activity [27]. The phytohormone Aux is an essential phytohormone regulator of apical dominance, cell elongation, primary root elongation, lateral root formation, plant embryogenesis, and many others [31,32,33]. The moderately short but universal Aux signalling pathway allows fast switching between repression and activation of gene transcription via auxin-dependent degradation of transcriptional repressors [33,34]. At high concentrations, it is assumed that auxin stabilises the interaction between TIR1/AFB and AUX/IAA transcriptional repressors, representing the core auxin signalling pathway together with auxin response factor (ARF) transcription factors [35]. When auxin concentrations decrease, the stability of AUX/IAA increases. Moreover, their interactions with ARF increase to repress their transcriptional activities [32]. It remains unclear how these relatively short auxin signalling cascades moderate many different cellular responses, including cell proliferation, expansion, and differentiation [32]. All of these presumably affect the activation or repression of completely different groups of genes and related subsequent cellular activities.

The results of the application of PEO-IAA on other plant species, such as *Arabidopsis thaliana*, showed that the use of the TIR1/AFBs auxin signalling pathway inhibitor (PEO-IAA) leads to chromatin loosening, indicating that auxin signalling functions to reduce chromatin accessibility [36]. It was proven that the transient reduction in auxin signalling by the application of PEO-IAA downregulated the expression of several fundamental cell cycle genes in *Arabidopsis* root meristem [32]. When *Arabidopsis* seedlings were treated with 10 μM PEO-IAA for 3 h, the endogenous IAA levels in the roots increased significantly compared with the control. Nevertheless, the difference was not significant in shoots. When seedlings were treated with PEO-IAA for a more extended period, endogenous IAA levels were induced not only in roots but also in shoots. Moreover, the expression of two genes from the *YUC* family was induced in the shoot after the presence of PEO-IAA. However, these results indicate that the influence of PEO-IAA is more significant in roots compared to shoots [37].

The method using PEO-IAA is already used in some research facilities [27], and when combined with the cytokinin derivative 6-benzylamino-9-(-tetrahydroxypyranyl) purine (BAP9THP), the successful formation of multiple shoot explant cultures has been achieved. When used in cultivar USO-31, which was also used in our research, this methodology resulted in a higher multiplication rate (isolated meristem = 3.6 ± 1.0; shoot apex = 2.8 ± 0.4; cotyledonary node = 1.4 ± 0.5) [27]. Nevertheless, the effect of the presence of PEO-IAA on gene expression and, consequently, on the production and the resulting concentration of cannabinoids was not investigated.

Following these findings, we studied the relative gene expression (RGE) of specific genes involved in the biosynthetic pathways of the major cannabinoids. Furthermore, we analysed the concentration levels of secondary metabolites in the dry weight (relative concentration (RC)) of plants cultivated on a medium containing PEO-IAA. We compared the RGE and RC results of the experimental variants with the control variants (without the addition of auxin inhibitor in the culture medium) to determine whether the presence of this substance in the culture medium affects the expression and concentration profiles examined, and, if so, whether the effect is positive or negative. The risk of an unfavourable influence on the gene expression and concentrations of pharmacologically essential secondary metabolites in the outcome plant material must be considered and carefully studied. This study is the first step toward obtaining essential data on this problem.

## 2. Results

This study focused on the changes in the relative gene expression profiles and changes in the concentrations of secondary metabolites (phytocannabinoids) in the presence of auxin antagonist PEO-IAA in two different cultivars of cannabis plants (‘USO-31’ and ‘Tatanka Pure CBD’). The genes of interest in our study, which encode enzymatic proteins involved in the cannabinoid metabolic pathway (*THCAS*, *CBDAS*, *CBCAS*, *OAC*), were investigated using the RT-qPCR method. The biosynthetic metabolic products (CBDA, CBD, CBCA, CBC, ∆^9^-THCA, ∆^9^-THC) were analysed via a chromatographic method.

### 2.1. Plant Material from the In Vitro Experiment

After the 4-week cultivation period, the experimental in vitro biomass was harvested. A detailed description of the in vitro experimental conditions is described in the Section 4.2. As shown in Figure 2 and Figure 3, the effect of PEO-IAA on the morphology of treated plants was more visible in the case of the ‘USO-31’ cultivar than in the case of the ‘Tatanka Pure CBD’ cultivar. Figure 2 and Figure 3 also show that plantlets with some level of hyperhydricity had been produced after the cultivation on media containing PEO-IAA. We discussed the reasons for this observation in more detail in the Section 3.

### 2.2. Relative Gene Expression (RGE)

The RT-qPCR method was used to measure gene expression changes in the presence of PEO-IAA in the cultivating media. The cDNAs for the analysis were obtained by transcription from RNAs isolated from four biological replicates of the in vitro experiment.

The result shows that all genes were expressed in both the control and the experimental variant, except for the *THCAS* gene, where the cycle of threshold values was below the detection limit. The rest of the RGEs profiles of the genes of interest were compared in each biological replicate between the control variant and the experimental variant that contained one micromolar concentration of PEO-IAA in its culture medium. The results of the average RGE of individual genes of interest are shown in the graphs (Figure 4).

The *OAC* gene was the only visible and significantly affected gene after averaging the values from individual biological replicates (Figure 4). It showed a 1.5-fold RGE decrease compared with the control in the ‘USO-31’ cultivar, and a 1.7-fold increase from the control in the ‘Tatanka Pure CBD’ cultivar. It should be noted that the values of the four biological replicates for the genotype ‘USO-31’ are listed neither in Table 1 nor in Figure 4. In this case, it was a defective sampling that produced unrealistic outlier values. Nevertheless, the results of the *t*-test (Table 1) do not prove any of these differences to be statistically significant.

### 2.3. Cannabinoid Content

The cannabinoid concentrations in samples of the cultivars ‘USO-31’ and ‘Tatanka Pure CBD’ were determined via ultra-high performance liquid chromatography coupled to a tandem mass spectrometer (UHPLC-MS/MS). Overall, the analysis showed higher concentrations of phytocannabinoids in the CBD genotype compared to the hemp cultivar ‘USO-31’ (Table 2). Moreover, quantifiable amounts of the neutral cannabinoids CBD, CBC, and the acid ∆^9^-THCA were found in this genotype.

Concerning the cannabinoid concentrations, there was a 1.7-fold decrease in CBDA in cultivar ‘USO-31’ compared to the control (Figure 5). For the cultivar ‘Tatanka Pure CBD’, the increases observed for the variant with ‘PEO-IAA’ for studied cannabinoids (CBCA, CBDA, and ∆^9^-THCA) were approximately 1.3 times higher than for the control variant (Figure 5). In the case of the cannabinoid ∆^9^-THCA, the acid was only detected in the cultivar ‘Tatanka Pure CBD’. Here, an approximately 1.47-fold increase was found for the PEO-IAA variant compared to the control. However, according to the results of the *t*-test (Table 3), statistical significance was proven only for the CBDA increase (*p*-value < 0.05).

### 2.4. The Effect of Gene Expression on Cannabinoid Concentration

We used regression analysis to investigate how changes in RGE (*CBCAS* and *CBDAS*) affect cannabinoid concentrations (CBCA and CBDA) for both tested cultivars. The values of relative concentrations (RCs) and RGEs (both in log-scale) were visualized in one scatter plot (Figure 6). The data suggest an increasing linear trend, although few outliers can be observed (namely in RGE and for Tatanka sample n. 2). We consider a regression model with ln(RC) set as a response variable and ln(RGE) as an explanatory variable, adjusted for the type of cannabinoid (C) and genotype (G). Table 4 reports the robust least trimmed squares (LTS) estimates of the regression parameters. Considering the usual 5% significance level threshold, all three variables are statistically significant. We can expect that for the given ln(RGE), the respective value of ln(RC) will be 1551 (1104; 1998) times higher, with the resulting value being increased by 0404 (0138; 0671) for CBDA and decreased by 0362 (0088; 0635) for Tatanka.

## 3. Discussion

In vitro cultivation and propagation of *Cannabis sativa* is a rapidly evolving field where many novel methodologies and techniques are being introduced. One essential part that needs improvement within cannabis tissue cultures is improving multiplication to speed up the process and increase production over time and input energy [38]. One of the strategies for this improvement is the chemical influence of apical dominance, for example, with the application of auxin antagonists [27]. The use of PEO-IAA as an inhibitor of apical dominance has already been proven by the Smýkalová et al.’s [27] team as a suitable approach to achieve successful multiple-shoot formation. Their research estimated that multiplication coefficients applying their protocol using PEO-IAA are up to 1:10. They also found that the newly formed shoots could be reliably rooted.

However, what remains unclear is whether the presence of PEO-IAA can negatively or positively affect the propagated material at the gene expression or the concentration levels of important secondary metabolites. It is known that the use of TIR1/AFBs auxin signalling pathway inhibitor (PEO-IAA) leads not only to the disruption of the Aux signalling pathway [27] but also to chromatin loosening [36]. Together, these two features can influence the activation or repression of entirely different gene groups and related cellular activities. Hasegawa et al. [36] found that PEO-IAA treatment significantly altered the expression of 3833 genes (*p* < 0.01): 1222 of them were down-regulated (PEO-IAA/DMSO ≤ 0.67-fold), and 1434 were up-regulated. The majority of studied genes were related to chromatin and chromosome organisation and methylations.

We focused on the leading essential genes associated with the cannabinoid metabolic pathway (*THCAS*, *CBDAS*, *CBCAS*, *OAC*) and the key secondary metabolites such as the most abundant cannabinoids (CBDA, CBD, CBCA, ∆^9^-THCA, ∆^9^-THC).

Since we worked only with young plants under in vitro conditions that were still in the vegetative stage of development, the measured RGE values of the studied genes were low, as were the relatively low levels of the concentrations of the studied cannabinoids. However, even under these circumstances, it was possible to compare the influence of the RGE rate on the resulting cannabinoid concentrations with the presence or absence of PEO-IAA. Unfortunately, RGE values of the *THCA* gene were below the limit of detection in all analysed samples. In addition, ∆^9^-THCA and ∆^9^-THC metabolite concentrations were below the detection limit in the USO-31 cultivar. The detection limits of the methods were achieved not only because of the work with young plants but mainly due to the choice of the cultivar, which does not exceed 1% ∆^9^-THC in mature, dry female inflorescences.

Applying statistical analyses detailed in the Section 2 and Section 4, we were able to confirm that the RGEs of the *CBCAS* and *CBDAS* genes were directly proportional to the RC of the cannabinoids CBCA and CBDA, respectively. A visible trend of this relationship is shown in Figure 6.

For all studied genes with RGEs above the detection limit (*OAC*, *CBCAS,* and *CBDAS*), no statistically significant differences in RGEs were observed between the variants, with the presence of PEO-IAA and control variants. This finding suggests that the gene expression of key genes related to the cannabinoid biosynthetic pathway is not affected by the presence of PEO-IAA.

Regarding the measured cannabinoid concentrations, minor decreases in the measured CBCA and CBDA values were observed in the cultivar ‘USO-31’ for the experimental variant with PEO-IAA. In contrast, slight increases in the concentrations of the three detected cannabinoids CBCA, CBDA, and ∆9-THCA were observed in the cultivar ‘Tatanka Pure CBD’ for the variant with PEO-IAA. However, a statistically significant difference was found only in the case of CBDA cannabinoid (at statistical significance level α = 0.05) in the ‘Tatanka Pure CBD’ cultivar. This statistically significant increase may not be due to an effect directly caused by a change in *CBDAS* gene expression, which we did not observe in this cultivar in the treated variant, but by a greater amount of some upstream metabolite in the biosynthetic pathway (e.g., olivetolic acid or CBGA).

As for the morphological changes between the control and experimental variants with the presence of PEO-IAA, they were more evident in the cultivar ‘USO-31’ than in the cultivar ‘Tatanka Pure CBD’. This observation may have been due to the genetic specifics of the cultivars. In field conditions, the Ukrainian cultivar hemp ‘USO-31’ reaches a height of up to 2.5 m, while the cultivar ‘Tatanka Pure CBD’ tends to be lower in outdoor conditions. Moreover, the hemp ‘USO-31’ cultivar, due to its genetic background (it comes predominantly from *Cannabis sativa* subsp. sativa), shows a greater degree of apical dominance than the cultivar ‘Tatanka Pure CBD’ (a hybrid variety with the genetic predominance of *Cannabis sativa* subsp. indica). Therefore, it can be assumed that the internal concentrations of auxins will be different in the two cultures, and therefore their response to the auxin antagonist may also differ.

One of the reasons why we have seen almost no changes in the level of gene expression and cannabinoid concentration may be the low concentration of PEO-IAA used. Although this concentration was confirmed as sufficient to derive the formation of multiple shoots [27], in other studies where fundamental changes in gene expression were observed, concentrations of up to 10-fold higher [32,36,37] or even 300-fold higher [39] were used. However, these studies investigated the auxin antagonist effect on *Arabidopsis thaliana*. Moreover, the plant material was exposed to these concentrations for a shorter period, as these studies were not aimed at increasing the multiplication coefficient of the shoots.

Another reason why we did not observe any statistically significant changes at the gene expression level may be because PEO-IAA exhibits an effect of auxin level alteration mainly in roots, as demonstrated in the research by Takato et al. [37]. Their experiments treated seedlings of Arabidopsis thaliana with 10 μM PEO-IAA for 3 h. The seedlings were harvested, and endogenous IAA levels were analysed in shoots and roots. Endogenous IAA levels were higher in the roots of PEO-IAA-treated seedlings than in mock-treated seedlings. Nevertheless, the difference was not significant in shoots [37].

Our findings suggest that using PEO-IAA for multiple shoot formation is an appropriate solution since its presence in the culture media most likely does not significantly disrupt the gene expression of known genes related to the biosynthetic pathway of the main cannabinoids, nor does it reduce cannabinoid concentrations. In one of the cases, the presence of PEO-IAA even increased the CBD content compared to the control. However, this increase would be unlikely to persist to a significant extent after transferring the explants to *ex vitro* conditions and during the transition of the plants to the generative flowering phase, which is essential for the production of cannabinoids.

## 4. Materials and Methods

### 4.1. Plant Material—Establishment of In Vitro Cultures

Two cultivars, a hemp cultivar ‘USO-31’ and a high-cannabidiol cultivar ‘Tatanka Pure CBD’, of *C. sativa* L. (*Cannabaceae*) were included in the present study. The cultivar ‘USO-31’ was kindly provided by the Czech Hemp Gene Bank (Agritec Ltd., Šumperk, Czech Republic), and the seed of the cultivar ‘Tatanka Pure CBD’ was purchased from the RQS Group Ltd. (Amsterdam, The Netherlands).

The in vitro cultures from the cultivar ‘USO-31’ were established by in vitro seed germination. The seeds were placed in a sterile beaker (100 mL) and surface-sterilized with 96% ethanol (*v*/*v*) by shaking at 135 rpm for 2 min. The seeds were then rinsed with sterile distilled water in an air-laminar box, and shaken with 10% sodium hypochlorite (*v*/*v*) and a drop of wetting agent (Tween20) for 30 min at laboratory temperature. Finally, the seeds were rinsed three times with sterile distilled water [27]. Seed germination was performed on solidified ½ strength MS medium including vitamins [40] (Duchefa Farma B.V., Haarlem, The Netherlands) (pH 5.8) supplemented with sucrose, 30 g·dm^−3^; plant agar, 8 g·dm^−3^; ascorbic acid, 0.02 g·dm^−3^; indole-3-butyric acid (IBA), 1 × 10^−5^ g·dm^−3^; 6-benzylaminopurine (BAP), 1 × 10^−5^ g·dm^−3^. The surface-sterilized seeds were germinated in 100 mL Erlenmeyer flasks containing 6 seeds each in a phytotron at 22 °C at 40% relative humidity for 3 days in darkness. The light regime was then changed to a 16/8 photoperiod (36 μmol m^−2^ s^−1^). Seedlings that had reached a height of 5–6 cm were transplanted to the 100 mL sterile Erlenmeyer flasks containing solidified culture medium with a composition as follows: ViVi 6 medium (Duchefa Farma B.V., Haarlem, Netherlands), 6.17 g·dm^−3^; sucrose, 30 g·dm^−3^; plant agar, 5.5 g·dm^−3^; ascorbic acid, 0.02 g·dm^−3^; thidiazuron (TDZ) (filter sterilized), 1 × 10^−4^ g·dm^−3^. The medium was also augmented with antibiotics (ampicillin, 0.133 g·dm^−3^; chloramphenicol, 0.066 g·dm^−3^) for controlling endogenic bacterial contamination. Antibiotics were dissolved in 0.002 dm^−3^ of dimethyl sulfoxide (DMSO) and filter sterilized. Cultivation of the ‘USO-31’ cultivar on ViVi 6 media was carried out for several months, with passage on fresh media every two weeks until the required amount of biomass for the further experiment was achieved (Figure 7).

The in vitro cultures from the ‘Tatanka Pure CBD’ cultivar were established by cuttings excised from healthy young *C. sativa* L. plants at the vegetative growth stage. The plant was propagated from seeds and cultivated in an indoor growth chamber at the Crop Research Institute (Olomouc, Czech Republic). The indoor conditions were as follows: the light regime was a 16/8 photoperiod (480 μmol·m^−2^·s^−1^), the humidity was 60%, and the temperature was 23/18 °C for light/dark. Nodal segments with a height of ca. 3 cm, containing axillary buds as well as one to two leaves from the healthy female mother (donor) plants, were used as cuttings. Cuttings were surface-sterilized in a sterile beaker (100 mL) with immersion in 70% ethanol (*v*/*v*) with continuous stirring for one minute, then shaken with 10% sodium hypochlorite (*v*/*v*) and a drop of wetting agent (Tween20) for 10 min at laboratory temperature. After surface sterilization, the cuttings were trimmed, and most of the expanded leaf area was removed. Three rinses followed each immersion with sterile distilled water. The surface sterilization process took place under sterile conditions of an air-laminar box, as did all the downstream handlings. Such sterile cuttings were placed in the 100 mL sterile Erlenmeyer flasks containing solidified ViVi 6 medium with the same composition as in the case of the cultivar ‘USO-31’. Cultivation of the ‘Tatanka Pure CBD’ cultivar on ViVi 6 media was carried out for several months, with passage on fresh media every two weeks until the required amount of biomass for the further experiment was achieved (Figure 8).

### 4.2. The In Vitro Experiment

After the required amount of biomass was achieved, the in vitro experiment was established. The establishment of the experiment was the same for both cultivars. The nodal segments were passaged on two types of hormone-free medium in 100 mL Erlenmeyer flasks—one containing 1 μM of PEO-IAA (marked as ‘+’) and the other without PEO-IAA, which served as a control (marked as ‘−’).

The composition of the ‘+’ medium was as follows: ViVi 6 medium (Duchefa Farma B.V., Haarlem, Netherlands), 6.17 g·dm^−3^; sucrose, 30 g·dm^−3^; plant agar, 5.5 g·dm^−3^; ascorbic acid, 0.02 g·dm^−3^; ampicillin, 0.133 g·dm^−3^; chloramphenicol, 0.066 g·dm^−3^; PPM, 0.001 dm^3^; PEO-IAA, 0.00029 g·dm^−3^. Antibiotics and PEO-IAA were dissolved in 0.002 dm^−3^ of dimethyl sulfoxide (DMSO) and filter sterilized. The composition of the ‘−’ medium was the same as that of the ‘+’ media, except the PEO-IAA was omitted.

Each Erlenmeyer flask contained five to six nodal segments growing in the standardized conditions in a phytotron at 22 °C at 40% relative humidity and the light regime of a 16/8 photoperiod (36 μmol·m^−2^·s^−1^) for 4 weeks (Figure 2 and Figure 3). After this growing period, the ‘+’ and ‘−’ biomass were separately harvested, collected into plastic Falcon tubes, and immediately frozen in liquid nitrogen. The representative samples of harvested biomass were used for the following RNA isolation, and the rest were used for the biochemical analysis. Before the RNA isolation and biochemical analysis, the harvested biomass was stored at −80 °C. The in vitro experiment was performed for each cultivar in three/four biological replications.

### 4.3. The RNA Isolation and Reverse Transcription to cDNA

Total RNA was isolated with a Spectrum Plant Total RNA Kit (Sigma-Aldrich, St. Louis, MO, USA). Homogenization of the frozen biomass was carried out using a frigid mortar and pestle with liquid nitrogen on dry ice. Representative quadruplicates were taken for RNA isolation from the homogenized biomass of each variant. The rest of the biomass was frozen with liquid nitrogen and lyophilized. Such lyophilized biomass was stored at −20 °C and then used for phytochemical analyses. The treatment with On-Column DNase I Digestion Set (Sigma-Aldrich, St. Louis, MO, USA) was performed during the isolation procedure. The check of concentrations and purity of isolated RNA were determined at A260/A280 and A260/A230 ratios using a NanoDrop 2000 spectrophotometer (ThermoScientific, Waltham, MA, USA). The quality of RNA and absence of genomic DNA were checked by agarose gel electrophoresis. Complementary cDNA was synthesized using the Sensi FAST cDNA synthesis kit (Meridian Bioscience Inc., Cincinnati, OH, USA) in a thermocycler Eppendorf Mastercycler Pro S vapo.protect (Eppendorf, Hamburg, Germany).

### 4.4. The RT-qPCR Analysis

Primers for all the genes of interest were taken from the article by Fulvio et al. [41], except for the primer for the olivetolic acid cyclase (OAC) gene, which was designed using the Primer3web version 4.1.0 online tool. The gene encoding actin (*ACT*) was chosen as the housekeeping gene. Primer sequences for this gene were used according to van Bakel et al. [42]. For detailed information on all the primers used, see Table 5.

The primers were firstly tested via standard PCR on genomic DNA from both cultivars in thermocycler Eppendorf Mastercycler Pro S vapo.protect (Eppendorf, Hamburg, Germany), and the PCR products were checked on agarose gel electrophoresis.

Quantitative real-time PCR was performed using SensiFAST SYBR No-ROX Kit (Meridian Bioscience Inc., Cincinnati, OH, USA) in a 96-well thermocycler CFX Connect Real-Time PCR Detection System (Bio-Rad Laboratories, Hercules, CA, USA). The expression levels of different sampling cycles were normalised with the Actin (Act) gene as a reference housekeeping gene [33]. Before the RT-qPCR analyses of RGE profiles of studied genes, primer pairs were subjected to annealing temperature optimization to determine the efficiency of the PCR reaction in the system we used. These efficiencies were further used in calculating RGE profiles using the Pfaffl method [43]. Reactions were performed with the following conditions: polymerase activation at 95 °C for 2 min, followed by 40 amplification cycles of 95 °C for 5 s, 60 °C for 10 s, and 72 °C for 20 s, after which a melting curve was analysed from 65 °C to 95 °C. The melting curve analysis verified the specificity of the PCR products.

### 4.5. Phytochemical Analysis

Phytocannabinoids were analysed from the lyophilized biomassvia UHPLC-MS/MS using an UltiMate™ 3000 UHPLC system (Thermo Fisher Scientific, Waltham, MA, USA) according to the previously reported methodology [44]. Briefly, 45 ± 1 mg of each sample was weighed in triplicates, followed by ethanolic extraction (96% ethanol, *v*/*v*). The sample extracts were sonicated for 15 min at laboratory temperature, centrifugated for 10 min (21.200× *g*, laboratory temperature), and filtrated (Nylon filters, 0.22 µm pore size, 13 mm diameter). Then, the supernatant was diluted with 70% acetonitrile containing 0.1% formic acid, and internal standards (CBD-d_3_, CBN-d_3_, ∆^9^-THC-d_3_) were added at this step.

The chromatographic separation was performed on a Luna Omega Polar C18 (100 × 2.1 mm; 1.6 µm particle size) UHPLC column (Phenomenex, Torrance, CA, USA), and the thermostated column compartment was set at 40 °C. A binary gradient started at 40% of mobile phase A (0.1% aqueous formic acid, *v*/*v*) and 60% of B (acetonitrile with 0.1% formic acid, *v*/*v*). Then, the mobile phase B linearly increased to 80% of B in 11 min, followed by an increase to 100% in 0.1 min. These conditions were held for 1.4 min. After that, the system was re-equilibrated at the initial conditions for 4.5 min. The flow rate was 0.3 mL/min, and the injection volume was 2 µL. The chromatographic instrument was coupled to a TSQ Quantum Access Max triple quadrupole mass spectrometer (Thermo Fisher Scientific, Waltham, MA, USA), and all cannabinoids were detected in positive ionization mode HESI+. The electrospray voltage was set to 3 kV and the capillary temperature to 320 °C.

The analyses were evaluated and processed using Xcalibur 1.2 software (Thermo Fisher Scientific). The phytocannabinoid standards of (-)-*trans*-∆^9^-tetrahydrocannabinol ((-)-*trans*-∆^9^-THC), (-)-*trans*-∆^9^-tetrahydrocannabinolic acid ((-)-*trans*-∆^9^-THCA-A), and (-)-*trans*-∆^9^-tetrahydrocannabinol-d_3_ ((-)-*trans*-∆^9^-THC-d_3_) were purchased from Lipomed (Arlesheim, Switzerland); cannabidiol-d_3_ (CBD-d_3_) from Cayman Chemical Europe (Tallinn, Estonia). The Cerilliant^®^ standards of cannabidiol (CBD), cannabidiolic acid (CBDA), cannabigerol (CBG), cannabigerolic acid (CBGA), cannabinol (CBN), cannabinolic acid (CBNA), cannabinol-d_3_ (CBN-d_3_), ∆^8^-tetrahydrocannabinol (∆^8^-THC), cannabichromene (CBC), cannabichromenic acid (CBCA), ∆^9^-tetrahydrocannabivarin (∆^9^-THCV), ∆^9^-tetrahydrocannabivarinic acid (∆^9^-THCVA), cannabidivarin (CBDV), cannabidivarinic acid (CBDVA), cannabicyclol (CBL), and cannabicyclolic acid (CBLA) were purchased from Merck (Darmstadt, Germany). Solvents and chemicals were from the following manufacturers: formic acid, Supelco^®^ LC-MS grade acetonitrile, and water (Merck, Darmstadt, Germany), and 96% ethanol (Lachner, Neratovice, Czechia).

### 4.6. Statistical Analyses

Statistical analysis was performed in RStudio (R Software version 4.1.0), using packages *ggplot2* and *robustbase*.

First, samples with PEO-IAA were compared with controls in terms of cannabinoids’ gene expression and content. ‘USO-31’ and ‘Tatanka Pure CBD’ samples were analysed separately. The relative differences in each of the 4 controls concerning the matching PEO-IAA sample were computed. A one-sample *t*-test was applied to each set of the log-transformed values of RGE, respectively. Relative content (RC) was also determined, testing if the mean value is different from 0 (i.e., if the effect of PEO-IAA is significant).

Next, we examined the relationship between RGE and RC of CBCA and CBDA. The regression model ln(RC) = β_0_ + β_1_ ln(RGE) + β_2_ C_CBDA_ +β_3_ G_Tatanka_ + ε was considered, where β_j_, j = 0, …, 3 are unknown regression coefficients, ε is the random error term, and C_CBDA_ and G_Tatanka_ are dummy variables indicating the cannabinoids, respectively, and genotype (0 for CBCA and 1 for CBDA, respectively, 0 for USO and 1 for Tatanka). Robust LTS (least trimmed squares) estimation of the regression parameters was performed.

## 5. Conclusions

In vitro cultivation and multiplication are becoming an essential part of the rocketing cannabis industry. More methods and approaches are being introduced to increase the multiplication coefficient and speed up the process of in vitro propagation. Still, little attention has been paid to how each method may affect the resulting plant material at the genetic level. According to the study of the relative expressions of genes associated with the cannabinoid biosynthetic pathway and cannabinoid concentrations, we have found that the presence of the auxin antagonist PEO-IAA did not significantly interfere with either of the two examined issues. Nevertheless, in our experiments, we observed a certain degree of hyperhydricity in some plantlets that grew on the PEO-IAA medium. This fact may be due to a disruption of auxin activity and should be considered a potential limitation of using auxin antagonists for in vitro multiplication. We researched two immensely different genotypes—hemp cultivar ‘USO-31’ and ‘Tatanka Pure CBD’, a high-CBD genotype. Therefore, we assume that the negative effect of the presence of one micromolar concentration of PEO-IAA in the culture media should not be a significant negative factor for other cultivars as well. However, further research on other genotypes is needed to understand the issue comprehensively. Moreover, additional monitoring of plants after the transfer to ex vitro conditions and reaching the generative flowering stage may also add valuable information to the effect of PEO-IAA on the main qualitative factors of the produced plant material.

## Figures and Tables

**Figure 1 plants-12-01664-f001:**
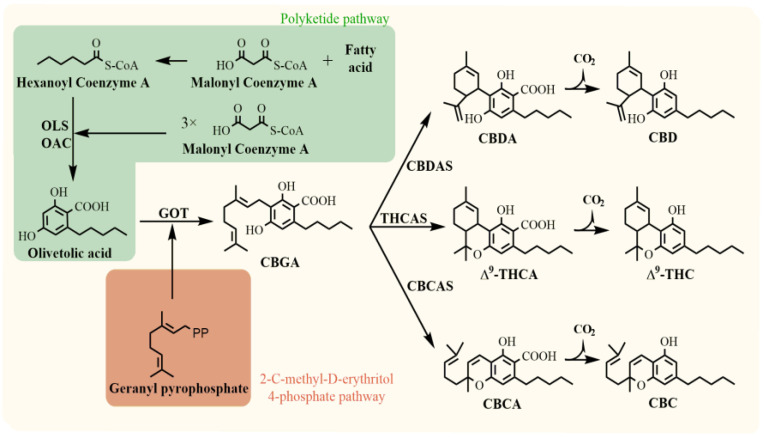
Biosynthesis of major phytocannabinoids. OLS, olivetol synthase; OAC, olivetolic acid cyclase; GOT, geranylpyrophosphate:olivetolate geranyltransferase; CBDAS, cannabidiolic acid synthase; THCAS, tetrahydrocannabinolic acid synthase; CBDAS, cannabichromenic acid synthase; CBGA, cannabigerolic acid; CBDA, cannabidiolic acid; ∆^9^-THCA, ∆^9^-tetrahydrocannabinolic acid; CBCA, cannabichromenic acid; CBD, cannabidiol; ∆^9^-THC, ∆^9^-tetrahydrocannabinol; CBC, cannabichromene [11,12].

**Figure 2 plants-12-01664-f002:**
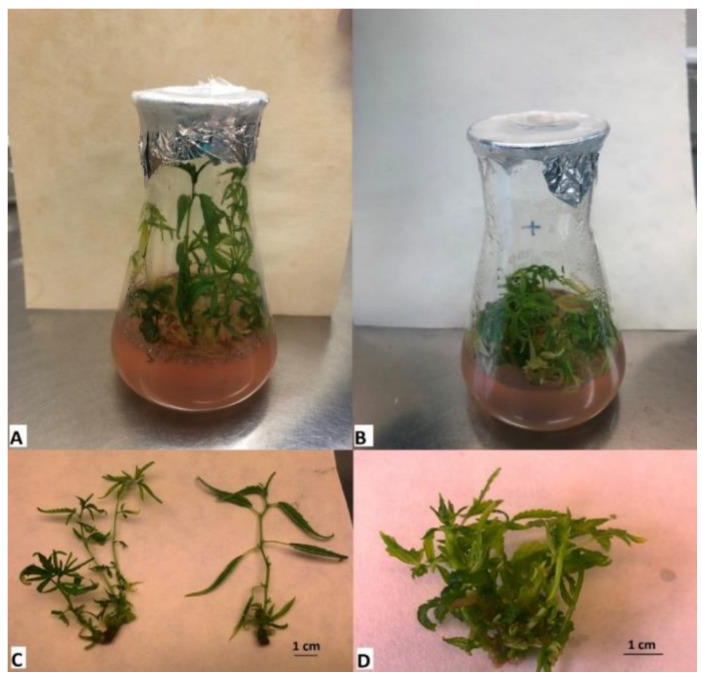
Comparison of experimental variants of the cultivar ‘USO-31’ after 4 weeks of cultivation in Erlenmeyer flasks. It is visible that the control variant showed greater apical dominance and elongating shoot growth (**A**,**C**), while the growing variant on the medium with the presence of PEO-IAA showed a lower growth with limited apical dominance and more developed lateral shoots (**B**,**D**).

**Figure 3 plants-12-01664-f003:**
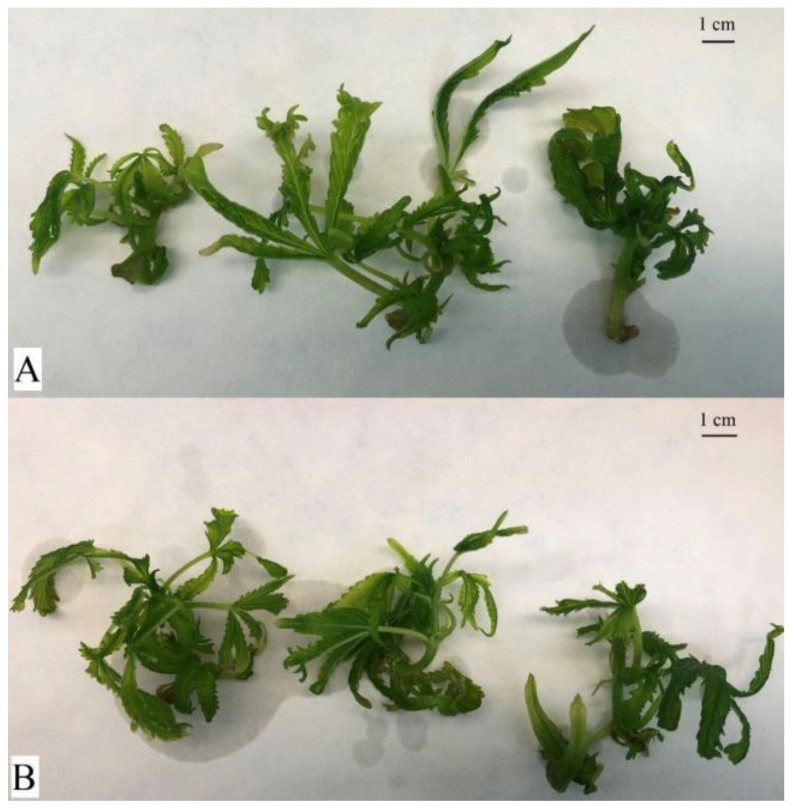
Comparison of experimental variants of the cultivar ‘Tatanka Pure CBD’ after 4 weeks of cultivation in Erlenmeyer flasks. Plant material at harvest before freezing with liquid nitrogen. The top shows the control variant (**A**). The bottom shows the experimental variant with the addition of PEO-IAA in the nutrient medium (**B**). The morphological differences between the variants are less visible in this cultivar than in the case of the ‘USO-31’ cultivar.

**Figure 4 plants-12-01664-f004:**
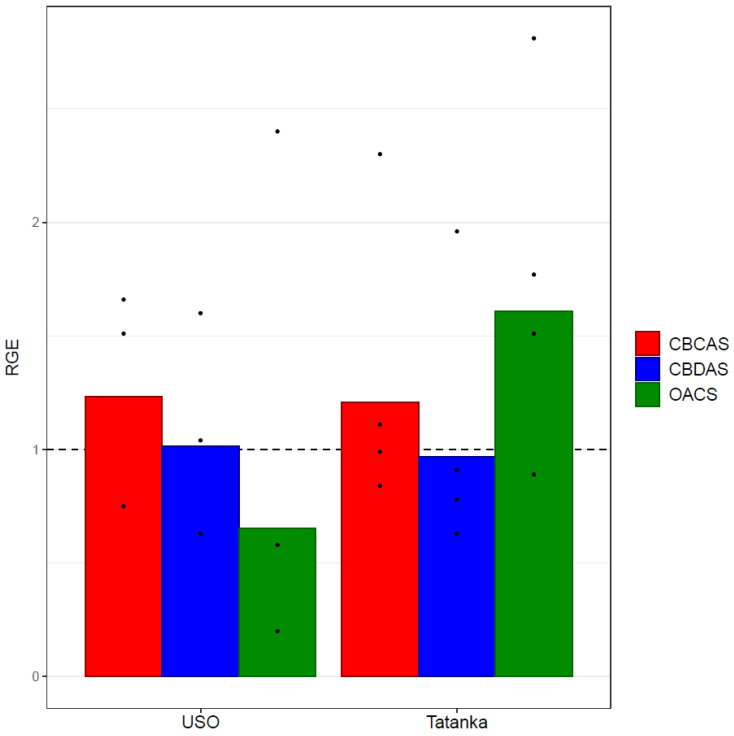
Bar plots show mean values of the given RGEs (*CBDAS*, *CBCAS*, and *OAC*) of the experimental variant *Cannabis sativa* ‘USO 31’ (without the outliers) and ‘Tatanka Pure CBD’ cultivars. The points represent the concrete values. Expression data were normalised using *Act* as a housekeeping gene, and the experimental variant was calibrated relative to the control variant (RGEs of the control variants = 1). No statistically significant differences (at statistical significance level α = 0.05) were found between the analysed variants.

**Figure 5 plants-12-01664-f005:**
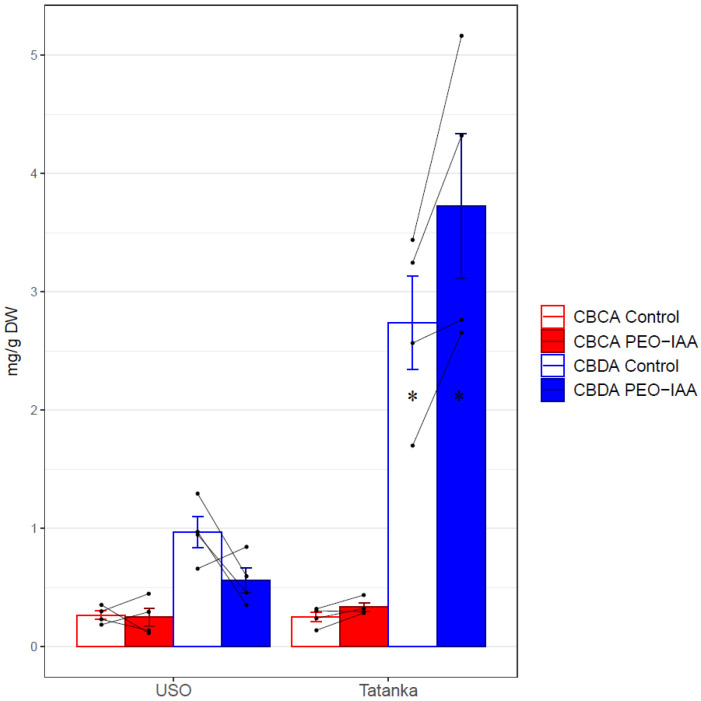
Bar plots show mean values and standard errors of the given concentration for both cultivars ‘USO-31’ and ‘Tatanka Pure CBD’. The points represent the concrete values. The matching samples are connected by lines. Significant differences in the concentrations of CBDA are represented by asterisks (*p* ≤ 0.05 “*”).

**Figure 6 plants-12-01664-f006:**
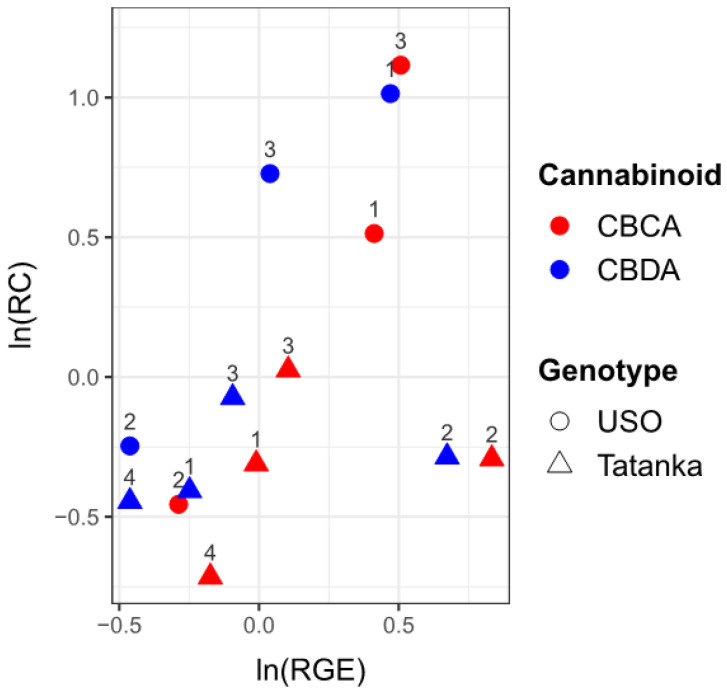
The values of ln(RC) are plotted against the values ln(RGE). The number above the points indicates the number of the given sample. A positive correlation between ‘RC’ values and ‘RGE’ values can be observed.

**Figure 7 plants-12-01664-f007:**
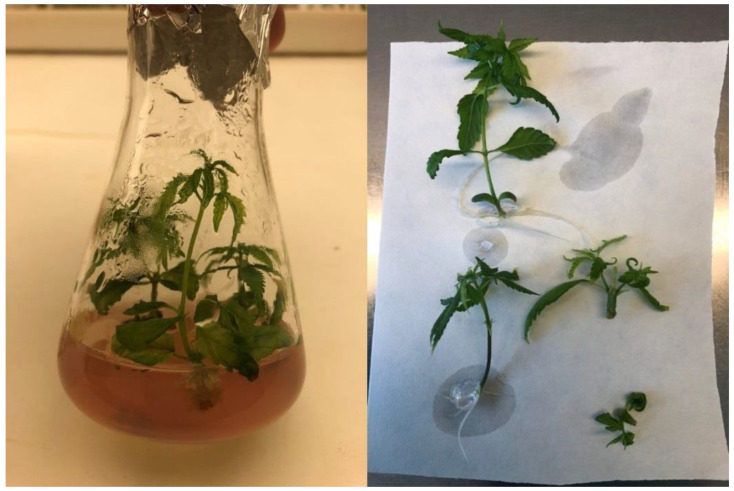
Multiplication of the ‘USO-31’ cultivar on ViVi 6 culture medium to achieve the required amount of biomass to establish the experiment with PEO-IAA.

**Figure 8 plants-12-01664-f008:**
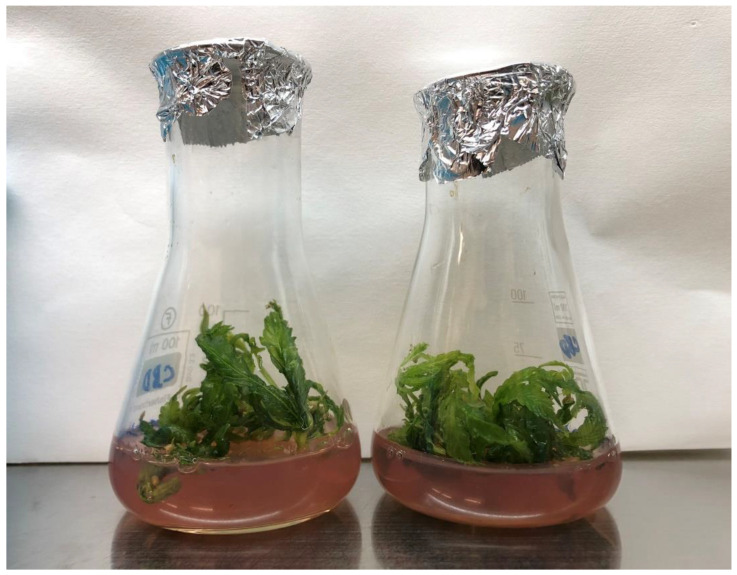
Multiplication of the ‘Tatanka Pure CBD’ cultivar on ViVi 6 culture medium to achieve the required amount of biomass to establish the experiment with PEO-IAA.

**Table 1 plants-12-01664-t001:** Results from the *t*-test applied on RGE for both cultivars.

ln(RGE)	USO-31			Tatanka Pure CBD		
Biological replicate	CBCA	CBDA	OAC	CBCA	CBDA	OAC
1	0.412	0.470	−0.545	−0.010	−0.248	0.571
2	−0.288	−0.462	0.875	0.833	0.673	1.033
3	0.507	0.039	−1.609	0.104	−0.094	0.412
4	NA	NA	NA	−0.174	−0.462	−0.117
*p*-value	0.489	0.959	0.614	0.460	0.902	0.139

**Table 2 plants-12-01664-t002:** Phytocannabinoid concentrations (mg/g DW) in samples of the cultivars ‘USO-31’ and ‘Tatanka Pure CBD’ and comparison of the content in the plants cultivated in the medium with PEO-IAA and control plants. All other cannabinoids not included in the table were in concentrations less than 0.009 mg/g DW (except for CBDVA (<0.03 mg/g), CBDA (<0.05 mg/g), CBGA (<0.18 mg/g), and CBC (<0.02 mg/g)). Values represent the means ± SD (n = 3).

Treatment	Cultivar	Biological Replicate	CBDA (mg/g DW *)	CBD (mg/g DW)	CBCA (mg/g DW)	∆^9^-THCA (mg/g DW)	∆^9^-THC (mg/g DW)
Control	USO-31	1	0.972 ± 0.016	<LLOQ	0.233 ± 0.019	<LLOQ	ND
	USO-31	2	0.661 ± 0.057	<LLOQ	0.188 ± 0.012	<LLOQ	ND
	USO-31	3	0.948 ± 0.076	<LLOQ	0.355 ± 0.020	<LLOQ	ND
	USO-31	4	1.295 ± 0.005	<LLOQ	0.301 ± 0.005	<LLOQ	ND
PEO-IAA	USO-31	1	0.353 ± 0.021	<LLOQ	0.139 ± 0.014	<LLOQ	ND
	USO-31	2	0.845 ± 0.099	<LLOQ	0.296 ± 0.011	<LLOQ	ND
	USO-31	3	0.458 ± 0.045	<LLOQ	0.116 ± 0.008	<LLOQ	ND
	USO-31	4	0.597 ± 0.049	<LLOQ	0.449 ± 0.031	<LLOQ	ND
Control	Tatanka Pure CBD	1	3.439 ± 0.124	0.083 ± 0.007	0.321 ± 0.006	0.113 ± 0.009	<LLOQ
	Tatanka Pure CBD	2	2.567 ± 0.266	0.089 ± 0.019	0.305 ± 0.003	0.064 ± 0.004	<LLOQ
	Tatanka Pure CBD	3	1.701 ± 0.121	0.066 ± 0.008	0.140 ± 0.001	0.053 ± 0.008	<LLOQ
	Tatanka Pure CBD	4	3.246 ± 0.174	0.111 ± 0.013	0.242 ± 0.009	0.111 ± 0.012	<LLOQ
PEO-IAA	Tatanka Pure CBD	1	5.164 ± 0.116	0.124 ± 0.015	0.438 ± 0.011	0.179 ± 0.007	0.009 ± 0.000
	Tatanka Pure CBD	2	2.763 ± 0.351	0.066 ± 0.012	0.297 ± 0.021	0.089 ± 0.007	<LLOQ
	Tatanka Pure CBD	3	2.654 ± 0.153	0.069 ± 0.006	0.286 ± 0.017	0.085 ± 0.003	<LLOQ
	Tatanka Pure CBD	4	4.319 ± 0.205	0.152 ± 0.012	0.324 ± 0.026	0.149 ± 0.018	0.009 ± 0.001

* DW, dry weight; ND, not detected; LLOQ, the lower limit of quantification (0.009 mg/g for CBD and ∆^9^-THC, 0.055 mg/g for ∆^9^-THCA).

**Table 3 plants-12-01664-t003:** Results from the *t*-test applied on RC in both cultivars. The *p*-value that denotes the statistical significance is in bold.

ln(RC)	USO-31		Tatanka Pure CBD		
Biological replicate	CBCA	CBDA	CBCA	CBDA	CBD
1	0.513	1.014	−0.311	−0.406	−0.405
2	−0.456	−0.246	−0.293	−0.285	−0.320
3	1.116	0.728	0.026	−0.074	−0.005
4	−0.400	0.774	−0.715	−0.445	−0.042
*p*-value	0.646	0.134	0.123	**0.036**	0.148

**Table 4 plants-12-01664-t004:** Regression coefficient estimates, standard errors, and *p*-values from LTS fit to cannabinoid relative concentration on relative gene expression (both in log-scale) with cannabinoid type and genotype as covariates. The *p*-values that denote the statistical significance are in bold.

Variable	Estimate	Std. Error	*p*-Value
Intercept	0.067	0.103	0.530
ln(RGE)	1.551	0.194	**<0.001**
C_CBDA_	0.404	0.115	**0.008**
G_Tatanka_	−0.362	0.119	**0.016**

**Table 5 plants-12-01664-t005:** List of primers for all studied genes. Forward (Fw) and reverse (Rv) primer sequences, amplicon length expressed in base pairs (bp), PCR efficiency (Eff), and annealing temperature (Ta).

Gene	Primer Sequence (5′–3′)	Amplicon Length (bp)	Efficiency (%)	Ta (°C)
*ACT*	Fw: CCAATAGCCTTGCATTCCAT	172	94.4	60
	Rw: TCGATTGGAAAGCCGAATAC			
*OAC*	Fw: TCATCCTGCCCATGTTGGAT	115	114.9	60
	Rv: AAGGCAGCTTGGTCGGCTAC			
*CBCAS*	Fw: GCTCACGACTCACTTCAGAACTAG	198	87.1	60
	Rv: GTAGAAGATGGTTGTATCAATCCAGCTC			
*CBDAS*	Fw: GCAATACACACTTACTTCTCTTCAGTTTTC	241	99.8	60
	Rv: ACGTAGTCTAACTTATCTTGAAAGCAC			
*THCAS*	Fw: AAAACTTCCTTAAATGCTTCTCAA	198	89.1	60
	Rv: TAAAATAGTTGCTTGGATATGGGAGTT			

## Data Availability

The data presented in this study are available on request from the corresponding author.

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
