# Peer review of "Investigation of the Effect of the Auxin Antagonist PEO-IAA on Cannabinoid Gene Expression and Content in Cannabis sativa L. Plants under In Vitro Conditions"

_plants, 2023, doi:10.3390/plants12081664_

Round 1
Reviewer 1 Report
In the current manuscript, ŠenkyÅ™ík et al. studied the relative gene expression (RGE) of specific genes involved in the biosynthetic pathways of the major cannabinoids. Furthermore, they analyzed the concentration levels of secondary metabolites in the dry weight of plants cultivated on a medium containing PEO-IAA. They also compared the results with the control group (no addition of the inhibitor to the culture medium) to confirm or refute the suitability of using this substance for enhancing the in vitro propagation of cannabis plant shoots since this method is being used in some research facilities and even in some commercial sectors. Although the topic is attractive, there are some concerns that should be addressed.
-Generally, the manuscript is well organized but has some typographical and grammatical errors.
-The paper title is well stated, and it is informative and concise.
-Abstract is well structured.
-The introduction was well written; however, it needs some improvements. The authors missed providing citations to support some sentences.
- Line 34-35: It is better, first, to introduce different applications of cannabis and then hemp and drug-type cannabis. So, "Historically, the plant....nutritionally rich seeds." should be changed to "Cannabis has been widely cultivated due to its industrial (DOI: 10.3906/bot-1907-15), nutritional (10.3390/plants11233330), medicinal, and recreational (10.1016/j.biotechadv.2022.108074) potentials. From regulatory and application perspectives, cannabis plants are categorized based on the level of Δ9-tetrahydrocannabinol (THC), one of the most important phytocannabinoids (10.1146/annurev-arplant-081519-040203). Plants are generally classified and regulated as industrial hemp if it contains less than 0.3 % THC in the dried flower (this level varies by country) or drug-type with more than this threshold (10.1016/j.indcrop.2020.113026)."
- Lines 87-89: light intensity and quality as well as carbohydrate sources also play fundamental roles in in vitro culture of cannabis. the authors should revise this sentence and include the importance of light and carbohydrate sources (10.3390/biology11050729; 10.3389/fpls.2021.757869)
L94-97: “Large-scale micropropagation….still understudied”. Support the sentence with appropriate citations related to the biotechnology of cannabis (10.3390/ijms22115671; 10.3389/fgeed.2022.823486; 10.3390/plants11091236)
-The results obtained in this study are interesting. The discussion is presented correctly. However, Based on Fig. 2 and 3, plantlets with high levels of hyperhydricity had been produced. The authors should present these observations in the result section.
- Material and research methods are presented appropriately. The experimental setup and the description in the methods section are well structured, and the statistical analysis is correctly performed.
- I suggest that the authors mention the limitations (e.g., produced plantlets with high levels of hyperhydricity) of the present study at the conclusion part and specify the follow-up tests.
Author Response
Dear reviewer,
Thank you for your review and comment.
- We have improved lines 34-35 in the 'Introduction' section as you suggested.
- In lines 87-89 of the 'Introduction' section, we added suggested references regarding the light intensity & quality and carbohydrate sources.
- In lines 94-97 of the 'Introduction' section, we added suggested references.
- We have added extra information about hyperhydricity to the 'Results' and 'Conclusion' sections.
Best regards,
Ing. Josef Baltazar ŠenkyÅ™ík et al.
Reviewer 2 Report
In general, it is a rather interesting research approach although without important innovation. My concern is regarding the plagiarism found which is in some publications from 2% which is rather high. I think that this is mostly in the introduction and the Material and methods which could be easily fixed from the writers. In total I saw a 34% similarity index (31% publications, 18% internet, 4% student papers). Otherwise, I would suggest that the paper could be published as it is. Now, I would prefer they see this plagiarism and also the references alphabetical order.
Author Response
Dear reviewer,
Thank you for your review and comment.
Regarding your concern about plagiarism, I believe it is a minor problem connecting mainly with the introduction, material, and method sections. After we send our corrected manuscript after correcting all of the issues addressed by all the reviewers, I believe the problem will disappear.
Regarding the order of references. We believe we made the reference list as is the norm for the Plants journal.
Best regards,
Ing. Josef Baltazar ŠenkyÅ™ík et al.
Reviewer 3 Report
Plants obtained from seeds are not uniform.
Have you cloned the individual genotypes?
Author Response
Dear reviewer,
Thank you for your review and comment.
We are aware that the plants from seeds are not uniform. Therefore, we used mixed and randomized groups of plants (which came from different seeds) for our experiments to filter out non-uniformity caused by slightly different genetic backgrounds. So the individual genotypes were clonally propagated and mixed into randomized groups.
Best regards,
Ing. Josef Baltazar ŠenkyÅ™ík et al.
Round 2
Reviewer 1 Report
All the comments have been addressed. I think that the current version of the manuscript can be published in Plants.
Author Response
Thank you for your approval of our manuscript for publication.